

# Nonequilibrium probability currents in optically-driven colloidal suspensions

Samudrajit Thapa[1,2,3], Daniel Zaretzky[4], Ron Vatash[4], Grzegorz Gradziuk[5], Chase Broedersz[5,6], Yair Shokef[1,2,7,8] and Yael Roichman[4,7,9⋆]

**1** School of Mechanical Engineering, Tel Aviv University, Tel Aviv 69978, Israel
**2** Center for Computational Molecular and Materials Science,
Tel Aviv University, Tel Aviv 69978, Israel
**3** Max Planck Institute for the Physics of Complex Systems,
Nöthnitzer Straße 38, 01187 Dresden, Germany
**4** School of Chemistry, Tel Aviv University, Tel Aviv 69978, Israel
**5** Arnold Sommerfeld Center for Theoretical Physics,
Ludwig Maximilians Universität München, Theresienstr. 37, 80333 Munich, Germany
**6** Department of Physics and Astronomy, Vrije Universiteit Amsterdam,
1081 HV Amsterdam, The Netherlands
**7** Center for Physics and Chemistry of Living Systems,
Tel Aviv University, 69978, Tel Aviv, Israel
**8** International Institute for Sustainability with Knotted Chiral Meta Matter,
Hiroshima University, Higashi-Hiroshima, Hiroshima 739-8526, Japan
**9** School of Physics & Astronomy, Tel Aviv University, Tel Aviv 69978, Israel

⋆ roichman@tauex.tau.ac.il

## Abstract

**In the absence of directional motion it is often hard to recognize athermal fluctuations. Probability currents provide such a measure in terms of the rate at which they enclose area in the reduced phase space. We measure this area enclosing rate for trapped colloidal particles, where only one particle is driven. By combining experiment, theory, and simulation, we single out the effect of the different time scales in the system on the measured probability currents. In this controlled experimental setup, particles interact hydrodynamically. These interactions lead to a strong spatial dependence of the probability currents and to a local influence of athermal agitation. In a multiple-particle system, we show that even when the driving acts only on one particle, probability currents occur between other, non-driven particles. This may have significant implications for the interpretation of fluctuations in biological systems containing elastic networks in addition to a suspending fluid.**



# 1  Introduction

How do you determine that a system is out of thermal equilibrium? Naturally, if you observe the system evolving in time or you see directional motion, the answer is trivial. However, if the system fluctuates around a steady state, it is not straightforward to distinguish between thermal and athermal fluctuations. For example, we know that living systems are far from thermal equilibrium, however, it may be hard to determine if some observed fluctuations in them stem from thermal noise or from biological activity [1–4]. There have been several approaches to address this issue in living systems [5–7] and in synthetic and biomimetic systems, such as in vibrated granular beds [8] and reconstituted biopolymer networks [9,10]. One approach is to look for violations of the fluctuation-dissipation theorem [1,5,6,11,12]. However, this entails not only measuring the spontaneous fluctuations in the steady state but also requires the application of some external perturbation to measure the system's non-equilibrium response.

Alternative non-invasive approaches based on stochastic thermodynamics search for entropy production or irreversibility in the fluctuations [13–17]. Here we build on these approaches, which allow, in a model-free manner to quantify deviations from equilibrium using any two measured degrees of freedom. Specifically, we consider nonequilibrium probability currents in a reduced phase space of the system. The phase space of a complex system is generally high dimensional, and includes the positions and momenta of all particles, yet one can

consider the projection onto a two-dimensional plane spanned by any two measurable quantities. During the system's temporal evolution, its trajectory in this reduced phase space will encircle an area. The rate at which this area increases – the area enclosing rate (AER) – serves as a measure to quantify nonequilibrium probability currents.

As proof of principle, this approach was applied to a simple theoretical model of two masses connected with springs and in contact with two different heat baths [4]. This approach has gained considerable attention via applications to biological [18, 19], climate [20] and electronic systems [21, 22]. However, the direct connection between the underlying activity in the system and its manifestation in the AER is not fully understood.

We use holographic optical tweezers to tune the nonequilibrium driving of a colloidal system, and measure the AER as a function of driving strength and interparticle separation. In this system, particles are coupled via long-ranged hydrodynamic interactions, and are driven by stochastic repositioning of the optical traps at a constant rate. Such stochastic repositioning of optical traps have been previously used to study Brownian particles in an active bath [23], heat fluxes between hydrodynamically interacting beads in optical traps [24, 25], and the conditions for the validation of a quasi fluctuation-dissipation theorem [12]. Using experiments, analytical theory and numerical simulations, here we show how hydrodynamic interactions give rise to algebraic scaling of the AER with particle separation. We relate the amplitude and rate of trap repositioning to the strength of nonequilibrium fluctuations in the system, and their subsequent effect on the AER. The dynamics of this system is governed by three time scales: the trap repositioning rate, the hydrodynamic relaxation rate, and the measurement rate. We show that the interplay between these scales is crucial for optimal observation of the probability currents, as measured by the AER. Finally, we show that in a multiple-particle system, even when the driving acts on one degree of freedom, probability currents occur also between other, non-driven degrees of freedom.

This article is organized as follows. After the introduction in Section 1, we present the experimental details in Section 2 and the details of numerical simulations in Section 3. In Section 4 we compare the results from numerical simulations with that from experiments for a system of two particles. Section 5 recalls the theoretical framework for calculating the AER starting with the Langevin equation while Section 6 extends the framework to include hydrodynamic interactions. In Section 7 we present results for the AER in case of a system of two hydrodynamically interacting particles where one of the particles is optically driven, and in Section 8 we extend our analysis to a system of three particles. Finally we discuss our results in Section 9.

## 2 Experimental design

Our experimental setup, schematically shown in Fig. 1a, consists of two or three colloidal particles (silica, diameter $d = 1.5 \pm 0.08 \ \mu$m) suspended in double distilled deionized water and trapped optically. We trap each particle in a separate optical trap and we independently and dynamically control the position of each trap with trap positioning precision of 10 nm. The motion of the colloidal particles in the experiments is three-dimensional, but we focus only on the one-dimensional projection of their motion along the line connecting the traps, which we define as the $x$-axis. See Ref. [26] for a theoretical study of the effects of the motion of two interacting beads on a plane. Each optical trap creates an effective potential that is usually approximated by a parabolic form, $U(x) = \frac{k}{2}(x - x_{\text{trap}})^2$, with $x_{\text{trap}}$ the position of the trap, $x$ the position of the particle, and $k$ the effective stiffness of the trap, which we control by modifying the laser intensity. In our setup, one of the particles is driven randomly by rapidly switching the location of its optical trap along the $x$-axis. That trap's new position is updated at regular

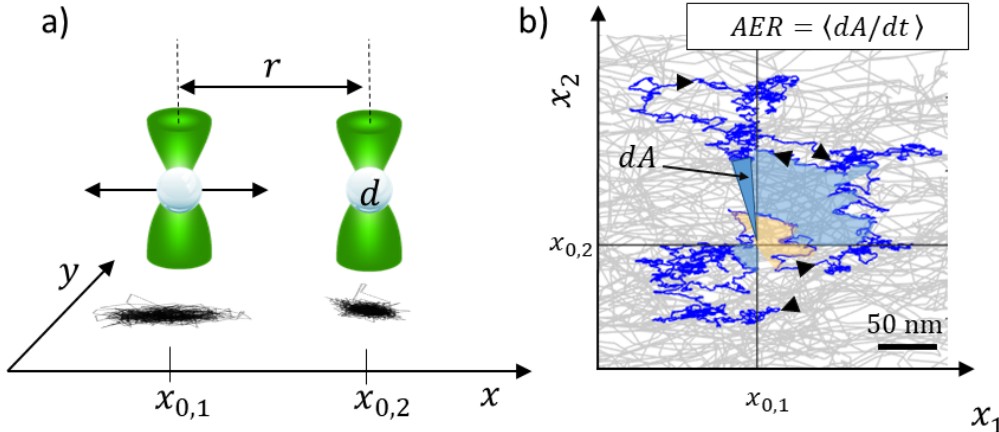

Figure 1: (a) Schematic illustration of our two-particle experimental setup. One particle is in a trap that remains at $x_{\text{trap},2} = x_{0,2}$ throughout the experiment. The second particle is driven by a trap with a position $x_{\text{trap},1}(t)$ that is regularly switched along the $x$-axis around an average position $x_{0,1}$. The trajectories of the particles are plotted below the traps. As a result of the driving, the trajectory of the driven particle is stretched along the $x$-axis. (b) The area enclosing rate (AER) is defined as the growth rate of the area enclosed by the trajectory in the phase space spanned by the $x$ motion of the two particles; $x_1(t)$ and $x_2(t)$. Here we show in gray a short portion (100 s) of the full phase-space trajectory, and in blue, the trajectory smoothed over a 25 s window, to highlight the average circulation in phase space. Areas swept in the counterclockwise direction are colored light blue and clockwise in orange. Experimental parameters are $r/d = 4$ and $b_0 = 73 \pm 10$ nm.

time intervals $\tau$ and drawn from a normal distribution, $p(x_{\text{trap}}) \sim \exp\left[-\left(x_{\text{trap}} - x_0\right)^2 / (2b_0^2)\right]$ centered at the particle's reference position $x_0$. We vary the nonequilibrium driving strength by changing the standard deviation $b_0$ in the driven trap's position distribution. The interaction between the particles is governed by the distance $r$ between the average positions of neighboring traps. Fig. 1b corresponds to a setup with $r = 4d$ and $b_0 = 73 \pm 10$ nm.

We use a home-built holographic optical tweezers setup [27–29] to project and switch the location of the optical traps. The setup is based on a continuous-wave laser operating at a wavelength of $\lambda = 532$ nm (Coherent, Verdi 6W) with a Gaussian beam profile. The laser beam is projected on to a spatial light modulator (Hamamatsu, LCOS-SLM, X10468-04) and is thus imprinted with a phase pattern. The beam is then relayed to the back aperture of a 100x oil immersion objective (NA 1.42) mounted on an Olympus IX 71 microscope. An optical trap is formed at a position prescribed by the phase pattern at the sample plane of the microscope. Switching the trap location is done by changing the phase pattern at a rate of $1/\tau = 36 \pm 1$ Hz.

The motion of the particles is recorded by a CMOS camera (FLIR, Grasshopper, GS3-U3-2356M) at 120 fps. We use conventional video microscopy [30] to extract the trajectories of the particles with 20 nm spatial resolution. To enhance the AER measurement, we trap particle 1 in a stiff trap that ensures its immediate response to the trap's displacement, while particle 2 is placed in a soft trap that allows a large displacement in response to a mechanical perturbation. In Fig. 2 the position distribution of both particles is compared between static conditions (blue) and when particle 1 is driven (orange). We obtain the effective stiffness $k$ of each trap by employing the equipartition theorem for the non-driven case, i.e. $\frac{1}{2}k\langle\Delta x^2\rangle = \frac{1}{2}k_B T$, where $k_B T$ is the average thermal energy at room temperature, and $\langle\Delta x^2\rangle$ is the measured variance of the position of each particle in its trap.

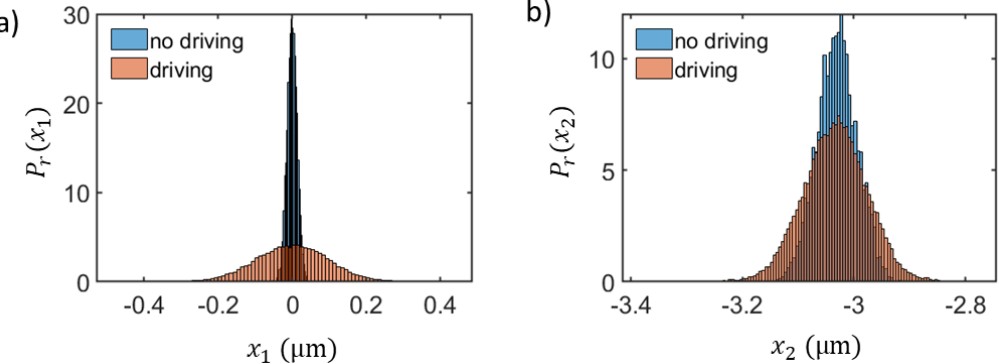

Figure 2: Position distribution within the traps of the driven particle 1 (a) and the non-driven particle 2 (b) in a two-particle experiment with driving strength $b_0 = 110 \pm 10$ nm and dimensionless trap separation $r/d = 4$. Both distributions are wider when particle 1 is driven (orange) than they are when both traps are static (blue).

We consider the phase space projected onto the two-dimensional space spanned by the $x$ displacements $\delta x_1 = x_1 - x_{0,1}$ and $\delta x_2 = x_2 - x_{0,2}$ of each particle from its mean trap position. Plotting the average two-dimensional probability density and current (Fig. 3a) we observe a non-vanishing probability current. We calculate the time-averaged AER from the dynamical trajectories by summing the triangular areas defined by every two consecutive points on the system's trajectory and the origin in this phase space (Fig. 1b), and dividing by the duration of the measurement. The area enclosed by the trajectory, $A_{12}$, has positive contributions for counterclockwise circulation and negative contributions for clockwise circulation. In Fig. 3b we show the evolution of the AER as a function of averaging time for several different experiments performed in the same conditions. Clearly, $A_{12}$ reaches a steady value after approximately 80 s. Hence, all our measurements of the AER include trajectories of at least this duration. It is difficult to infer the plateau values from the main panel of Fig. 3b. Therefore, the figure includes an inset which zooms in on the long-time behavior, and clearly shows that the plateau value is non-zero. From the distribution of the steady state values that we obtain (see Fig. 3b, inset), we estimate the error of our measurement of the AER to be 10 nm$^2$/ms. This is also the value of $A_{12}$ that we measure for systems with no driving.

## 3 Description of numerical simulations

In this section we briefly describe the details of numerical simulations, before we present the comparison of our experimental results with extensive two-dimensional Stokesian dynamics simulations [31] in the following section. This simulation protocol is well suited to calculate the thermal motion of many particles subjected to external forces and interacting via hydrodynamic interactions and hard-core repulsion [29]. Our simulations consider two-dimensional motion, and use the Rotne–Prager approximation [32] for the hydrodynamic interactions between the particles, as given by Eq. (14) below. The trap repositioning is done only for particle 1 and only along the $x$-axis which connects the particles. The simulations were performed with a simulation time step of $10^{-5}$ s for typical durations of 600 s. Similar to the experiments, we used a diameter $d = 1.5\ \mu$m for the particles, and the dynamic viscosity of water is given by $\eta = 0.89 \times 10^{-3}$ Pa·s. The homemade simulation code is based on previous simulations [29], and is publicly available [33].

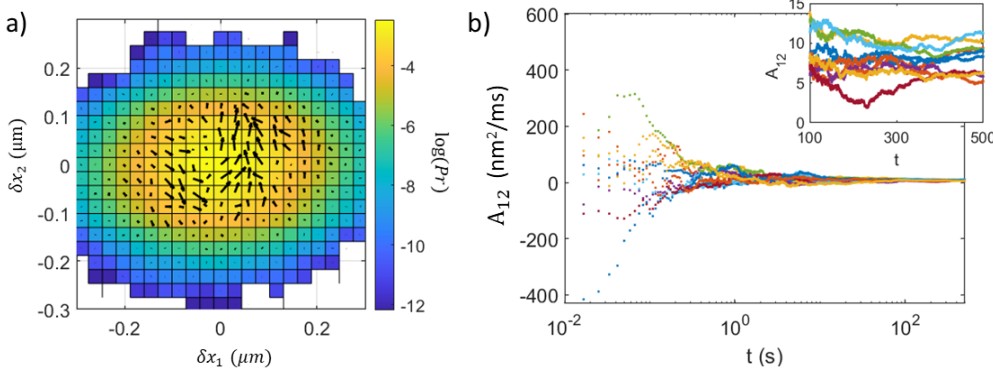

Figure 3: Probability currents in two-particle experiments with $b_0 = 110 \pm 10$ nm and $r/d = 4$. (a) phase space projected onto the two-dimensional space spanned by the displacements $\delta x_1 = x_1 - x_{0,1}$ and $\delta x_2 = x_2 - x_{0,2}$ of the two particles from their corresponding mean trap positions, color coded for probability density. The arrows indicate probability currents. The data is taken by averaging over 13 experiments, each of duration $\sim$500 s, hence a total of 780,000 frames, (b) The AER of several experiments in the same conditions, plotted vs averaging time. The inset shows a close up view of the AER at long averaging times, showing the variations between repetitions of the experiment and highlights that the AER for all the experiments saturates at non-zero values.

# 4 Comparison of results from experiments and numerical simulations

In this section we present the comparison of experimental results with those from numerical simulations. The minimal system exhibiting nonequilibrium probability currents requires two degrees of freedom. Thus, we consider the one-dimensional motion of two colloidal particles, optically trapped and driven as described above. We note that this system is reminiscent of the mass-spring model considered in [13] and discussed below. However, here particles influence one another via hydrodynamic interactions, and they are driven by the colored noise resulting from the stochastic trap repositioning at regular time intervals.

Figure 4 shows results from simulations and experiments for a fixed average distance of $r = 2d$ between the traps, and varying driving amplitudes $b_0$. The trap stiffnesses obtained from the experiments and used in the simulations were $k_1 = 2$ pN/$\mu$m, $k_2 = 0.5$ pN/$\mu$m. As seen in the figure, the experiments and simulations indicate a $b_0^2$ scaling of the AER with the driving amplitude. For weaker driving, the experimental AER is below the noise level, which we estimate from experiments without driving. Simulations predict AER which is higher by a factor of 5-10 compared to experiments.

The simulations and experiments presented in Fig. 5 show a $1/r$ decay of the AER with the distance $r$ between the traps. Similar to Fig. 4, we again see that the simulations give larger AER than the experiments. For each trap separation $r$ we measure somewhat different trap stiffnesses, and we show here results of simulations, in which we used traps with stiffnesses same as in Fig. 4. Note that the simulations presented here use the same measurement frequency of 120 fps as the experiments, in order to properly describe all the time scales in the experiments, even though the numerical time step in the simulations is much smaller.

We suggest that proximity of the particles to the boundary walls could explain the smaller values of AER observed in the experiments. The experiments were performed at an estimated distance of $\sim 2$ $\mu$m from the bottom wall while the height of the sample cell was $\sim 20$ $\mu$m,

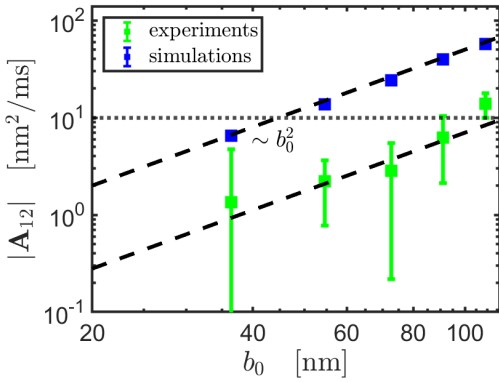

Figure 4: AER vs. driving strength for two particles separated by a distance of $r = 2d$ in experiments (green) and simulations (blue). The horizontal dotted line indicates the noise level, obtained from the AER measured in experiments without driving, and the dashed lines are guides to the eyes showing the $b_0^2$ scaling of AER with driving strength.

and the distances between the spheres were $3 - 8 \, \mu$m. Under these conditions, momentum is absorbed by both the bottom and top glass walls [34]. Indeed, the presence of walls leads to weaker hydrodynamic interactions between the particles, and causes it to decay faster with the distance between them as compared to the case without walls [35–37]. Thus, the motion of each particle due to the motion of other particles is smaller, and this results in a lower AER. However, because of the noise in the experimental data, the lack of precise information on the distance to the walls and the limited range of distance between the particles which can be explored, we are unable to provide a quantitative description of the effect of the walls. In order to check for other sources which might lower the AER in the experiments, we ran simulations replacing the parabolic traps with Gaussian traps mimicking the experiments, however, this resulted in AER values very close to the ones obtained with parabolic traps. We also performed simulations considering the effect of the size and spherical shape of the particles [38], however these too could not explain the discrepancy.

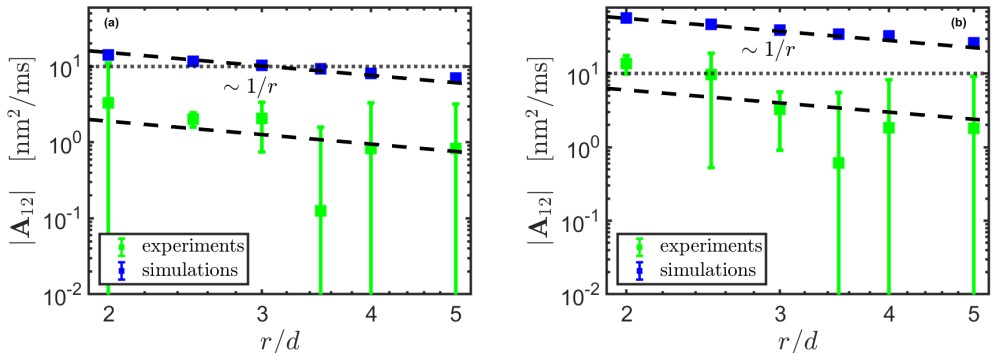

Figure 5: AER vs. the normalized distance between two particles for driving strengths $b_0 = 55$ nm (a) and $b_0 = 110$ nm (b), in experiments (green) and simulations (blue). The horizontal dotted line indicates the noise level, obtained from the AER measured in experiments without driving, and the dashed lines are guides to the eyes showing the $1/r$ scaling of AER with distance.

In the following sections, we present theoretical analysis explaining the $1/r$ decay of AER with distance and its $b_0^2$ increase with driving amplitude. This analysis will allow us to reveal the relations between the different time scales in the system and their effect on the AER. We will also discuss three-particle systems, where one particle is driven, and non-zero AER exists also in the phase space defined by the two non-driven particles. In our experimental system, this AER with three particles is below the noise level, but we clearly observe it in simulations.

## 5 Theoretical framework – From Langevin equation to AER

In this section we will recall previous results on how to compute the AER starting with a Langevin equation driven by white noise [21, 39]. This will set up the framework we subsequently use to obtain analytical results of the AER. We will also recall previous results on the AER for a mass-spring model [4, 13] and highlight that the theoretical results for this system are incapable of explaining the experimental results presented in Section 3. Consider a system with $N$ degrees of freedom, which evolves with time according to the following Langevin equation of motion

$$\frac{d\vec{\delta x}}{dt} = \mathbf{V}\vec{\delta x} + \mathbf{F}\vec{\xi},\tag{1}$$

with the column vectors

$$\vec{x} = \begin{pmatrix} x_1 \\ x_2 \\ \vdots \\ x_N \end{pmatrix}, \quad \vec{\delta x} = \begin{pmatrix} \delta x_1 \\ \delta x_2 \\ \vdots \\ \delta x_N \end{pmatrix}, \quad \vec{\xi} = \begin{pmatrix} \xi_1 \\ \xi_2 \\ \vdots \\ \xi_N \end{pmatrix},\tag{2}$$

denoting all the coordinates, their deviations from their equilibrium positions, and uncorrelated Gaussian white noise with unity variance, namely $\langle \xi_i(t)\xi_j(t') \rangle = \delta_{ij}\delta(t-t')$. The $N \times N$ matrix $\mathbf{V}$ captures the deterministic dynamics, and the $N \times N$ matrix $\mathbf{F}$ provides the amplitude of the noise. Equation (1) allows each of the different noise terms to act on all coordinates. Note that at first we present the analysis of the AER assuming the noise is white, and originates from thermal fluctuations, while the driving in our experiments and simulations contains a characteristic time scale $\tau$ of trap repositioning. In Section 7 we present the analysis of AER taking into account the colored nature of the noise [40]. We refer the interested reader to Ref. [41] for studies of the nonequilibrium steady-state distribution of the position of a damped particle confined in a harmonic trapping potential and experiencing active noise with short-time correlations. In Eq. (1) we may consider $\vec{x}$ to contain all degrees of freedom of the system. Then for $n$ particles in $d$ dimensions, the dimension of all vectors and matrices above is $N = nd$.

The Langevin equation (1) corresponds to the following Fokker-Planck equation, which gives the time evolution of the probability density $\rho(\vec{\delta x}, t)$ of the system,

$$\frac{d\rho(\vec{\delta x}, t)}{dt} = \nabla \cdot [\mathbf{V}\vec{\delta x}\rho(\vec{\delta x}, t)] + \nabla \cdot \mathbf{D}\nabla\rho(\vec{\delta x}, t),\tag{3}$$

where

$$\mathbf{D} = \frac{1}{2}\mathbf{F}\mathbf{F}^{\mathrm{T}},\tag{4}$$

is the diffusion matrix, and the superscript T denotes the transpose of a matrix. The steady-state solution is a Gaussian distribution with covariance matrix $\mathbf{C}$ obtained by solving the Lyapunov equation [4]

$$\mathbf{V}\mathbf{C} + \mathbf{C}\mathbf{V}^{\mathrm{T}} = -2\mathbf{D}.\tag{5}$$

The mean AER in the phase space projection spanned by $\delta x_i$ and $\delta x_j$ is then given by the $(i, j)$ element of the matrix $\mathbf{A}$, which is given by [21, 39],

$$\mathbf{A} = \frac{1}{2} \left( \mathbf{VC} - \mathbf{CV}^{\mathrm{T}} \right) . \tag{6}$$

Note that $\mathbf{A}$ is antisymmetric, $A_{ij} = -A_{ji}$, and the diagonal elements of $\mathbf{A}$ are trivially zero.

While non-zero AER is a signature of broken detailed balance and therefore the non-equilibrium nature of the system, the necessary and sufficient condition for detailed balance to be broken is [42]

$$\mathbf{B} = \mathbf{VD} - (\mathbf{VD})^{\mathrm{T}} \neq \mathbf{0} . \tag{7}$$

In case of a system at equilibrium with a symmetric matrix $\mathbf{V}$ together with a diagonal matrix $\mathbf{D}$ with identical diagonal elements—i.e. all the particles are at the same temperature—$\mathbf{B} = \mathbf{0}$, and therefore detailed balance is satisfied.

To demonstrate how this general framework is employed, we consider the simple mass-spring system [4, 13] schematically shown in Fig. 6, where particle 1 is in contact with a heat bath at temperature $T + \Delta T$, which is different from the temperature $T$ of the heat bath that particle 2 is in contact with. The particles themselves are connected to each other and to rigid walls at the ends via springs with stiffnesses $k_j$ as shown in the figure.

The deterministic response matrix $\mathbf{V}$ is given by

$$\mathbf{V} = \frac{1}{\gamma} \begin{pmatrix} -(k_1 + k_2) & k_2 \\ k_2 & -(k_2 + k_3) \end{pmatrix} , \tag{8}$$

with $\gamma$ the friction coefficient. The diffusion matrix is

$$\mathbf{D} = \frac{k_B}{\gamma} \begin{pmatrix} T + \Delta T & 0 \\ 0 & T \end{pmatrix} . \tag{9}$$

The noise matrix $\mathbf{F}$ is obtained by the Cholesky decomposition of $\mathbf{D} = \frac{1}{2} \mathbf{FF}^{\mathrm{T}}$ as

$$\mathbf{F} = \sqrt{\frac{2k_B}{\gamma}} \begin{pmatrix} \sqrt{T + \Delta T} & 0 \\ 0 & \sqrt{T} \end{pmatrix} . \tag{10}$$

We note that there are several ways to decompose $\mathbf{D}$ but the final result in terms of AER does not depend on which decomposition is used. The Cholesky decomposition is widely used because of the important property that the existence of Cholesky decomposition of a matrix means that the matrix is positive definite which ensures that the eigenvalues are positive. In the case of the diffusion matrix, this ensures that the diffusion coefficients are non-negative.

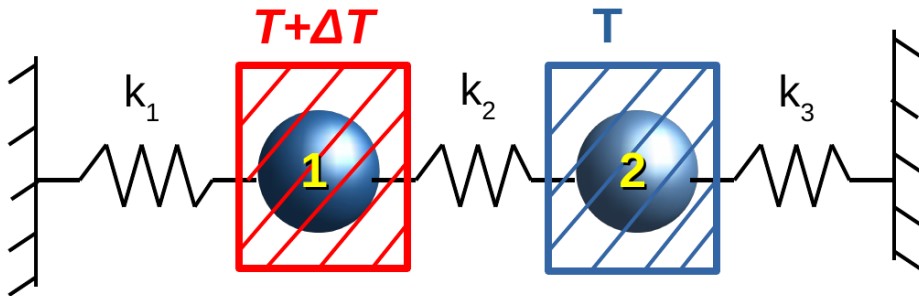

Figure 6: Schematic diagram of two particles connected with springs.

The matrix $\mathbf{B}$ is obtained as

$$\mathbf{B} = \frac{k_2 k_B \Delta T}{\gamma^2} \begin{pmatrix} 0 & -1 \\ 1 & 0 \end{pmatrix}. \tag{11}$$

Solving Eq. (5) gives the covariance matrix, the elements of which are

$$\mathbf{C}_{11} = \frac{k_B}{Z_1} \left[ T(2k_2^2 + k_3^2 + k_1 k_2 + k_1 k_3 + 3k_2 k_3) + \Delta T(k_2^2 + k_3^2 + k_1 k_2 + k_1 k_3 + 3k_2 k_3 \right], \tag{12a}$$

$$\mathbf{C}_{12} = \mathbf{C}_{21} = \frac{k_B k_2}{Z_1} \left[ T(k_1 + 2k_2 + k_3) + \Delta T(k_2 + k_3) \right], \tag{12b}$$

$$\mathbf{C}_{22} = \frac{k_B}{Z_1} \left[ T(k_1^2 + 2k_2^2 + 3k_1 k_2 + k_1 k_3 + k_2 k_3) + \Delta T k_2^2 \right], \tag{12c}$$

with $Z_1 = 2k_1 k_2^2 + k_1^2 k_2 + k_1 k_3^2 + k_1^2 k_3 + k_2 k_3^2 + 2k_2^2 k_3 + 4k_1 k_2 k_3$.

Finally the AER is obtained from Eq. (6) as [4]

$$\mathbf{A} = \frac{k_2 k_B \Delta T}{\gamma(k_1 + 2k_2 + k_3)} \begin{pmatrix} 0 & 1 \\ -1 & 0 \end{pmatrix}. \tag{13}$$

The AER scales linearly with the temperature difference $\Delta T$, and for identical springs $k_1 = k_2 = k_3$, it does not depend on the stiffness of the springs. We present extension of these results to a system of three particles in Appendix B. Note that the AER is independent of the distance between the particles, which does not enter the equations of motion. The mass-spring model therefore cannot be used to explain the experimental system we have because in our system we observe distance dependence of the AER as seen in Fig. 5. We note that, however, for a heterogeneously driven large elastic network of beads, tracking a pair of beads can result in measures of broken detailed balance that scale as a power law with the distance between beads [43].

## 6 Theory for trapped colloids suspended in fluid

In order to account for the distance dependence of the AER, which could not be explained by the mass-spring model in the previous section, in this section we consider a system of $n$ spherical particles interacting hydrodynamically in a liquid with drag coefficient $\gamma = 3\pi d\eta$, where $\eta$ is the dynamic viscocity of the liquid. Similarly to the analysis in the previous section, also here we will assume for now that the system is driven by white noise. To use the general prescription presented above for calculating the AER, we need to identify the matrices $\mathbf{V}$ and $\mathbf{F}$ entering the Langevin equation (1). For this physical system, the hydrodynamic interactions can be calculated within the Rotne–Prager approximation [32] that is suitable for well separated particles. The hydrodynamic interaction tensor is then given by [29, 32]

$$\mathbf{R}_{ij}^{\alpha\beta} = \begin{cases} \dfrac{\delta_{\alpha\beta}}{\gamma}, & \text{if } i = j, \\[2mm] \dfrac{3d}{8\gamma r_{ij}} \left( \delta_{\alpha\beta} + \dfrac{r_{ij}^\alpha r_{ij}^\beta}{r_{ij}^2} \right) + \dfrac{d^3}{16\gamma r_{ij}^3} \left( \delta_{\alpha\beta} - 3\dfrac{r_{ij}^\alpha r_{ij}^\beta}{r_{ij}^2} \right), & \text{if } i \neq j, \end{cases} \tag{14}$$

where $i, j$ are indices referring to particles, and $\alpha, \beta$ denote spatial coordinates. The diameter of the particles is $d$, while $r_{ij}$ denotes the distance between particles $i$ and $j$. The tensor $R_{ij}^{\alpha\beta}$ serves as a mobility tensor, namely, if a force $f_j^\beta$ is applied on particle $j$ in direction $\beta$, the resulting velocity of particle $i$ in direction $\alpha$ is $v_i^\alpha = R_{ij}^{\alpha\beta} f_j^\beta$. As shown below, this enters both

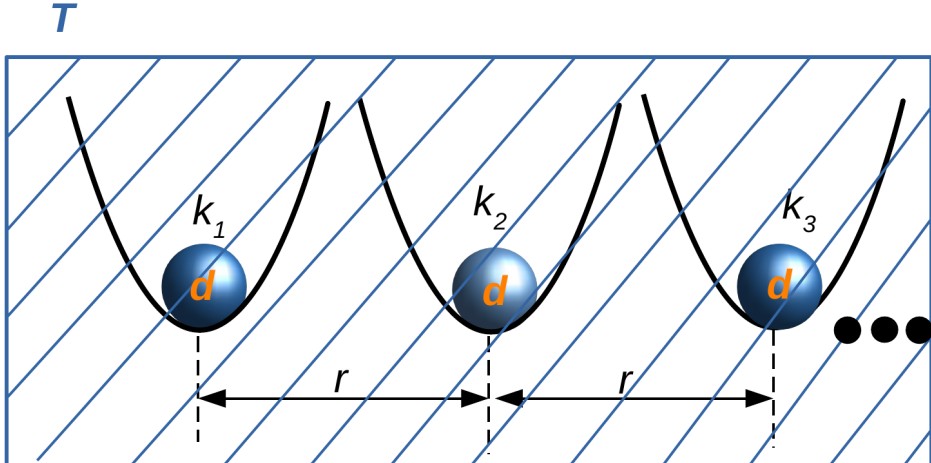

Figure 7: Schematic diagram of a one-dimensional array of particles in harmonic potentials.

the deterministic response matrix **V** and the noise matrix **F**. All our results from simulations are from two-dimensional motion using the Rotne Prager tensor given by Eq. (14). The choice of the Rotne-Prager tensor ensures momentum conservation in the simulations. Our analytical results however consider the Oseen tensor which keeps only terms up to order $1/r$ in the interaction tensor. The good agreement of analytical and simulation results presented in the subsequent sections validate that considering interaction tensor up to order $1/r$ is sufficient for the studies presented here. We also verified that simulations using interaction terms up to order $1/r$ are sufficient.

In our analytical derivations, we consider the effects of hydrodynamic interactions on the one-dimensional motion of particles along the direction between them. The particles are in harmonic potentials of generally different stiffnesses $k_i$. The equilibrium positions of the particles are separated by a distance $r$. The schematic for such a system of particles is shown in Fig. 7. For this one-dimensional situation, the Rotne-Prager tensor for hydrodynamic interaction reduces from Eq. (14) to $\mathbf{R} = \mathbf{H}/\gamma$, where the elements of **H** are given by

$$\mathbf{H}_{ij} = \begin{cases} 1, & \text{if } i = j, \\ \frac{3d}{4r_{ij}}, & \text{if } i \neq j, \end{cases} \tag{15}$$

and we have kept terms only up to order $1/r$.

Here we first consider the exactly solvable situation, in which particle 1 is in contact with a heat bath at temperature $T + \Delta T$ while the other particle(s) are in contact with a heat bath at temperature $T$, as depicted for two particles in Fig. 8a. Subsequently, in Section 7, we will consider the experimental situation, depicted for two particles in Fig. 8b, in which all the particles are in contact with a heat bath at ambient temperature $T$, and the trap of particle 1 is regularly repositioned, according to the experimental protocol described in Section 2. The former case corresponds to a Langevin equation driven by white noise for which we use the analytical expressions of the AER presented above, while the latter case corresponds to a Langevin equation with colored noise, for which we use the prescription of Ref. [40] to analytically calculate the AER. The two-temperature and the colored noise are separate out-of-equilibrium issues. We present the two-temperature case as an example to compare between the mass-spring system and the system of optically trapped particles. The colored noise case, on the other hand, mimics the experimental system where the trap position of one of the particles is repositioned periodically thereby driving the system out of equilibrium.

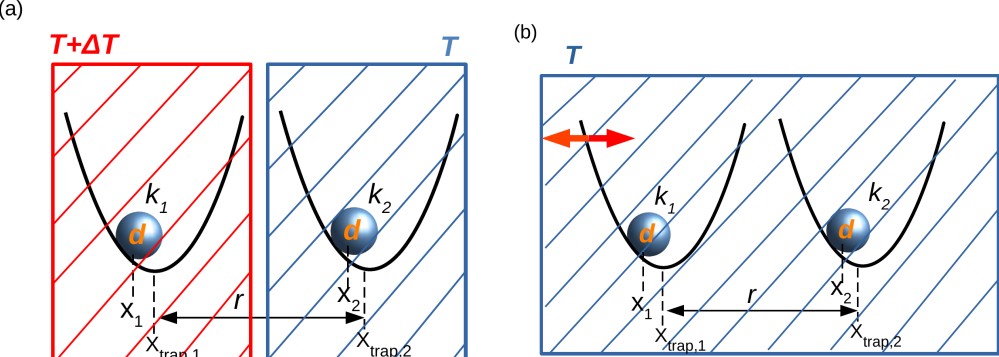

Figure 8: Schematic diagram of two particles in harmonic traps. (a) particle 1 is in contact with a heat bath at temperature $T + \Delta T$ and particle 2 is in contact with a heat bath at temperature $T$. (b) both particles are in contact with a heat bath at temperature $T$, and the trap of particle 1 is stochastically repositioned.

For the white-noise case, the dynamics is given by Eq. (1) with the drift matrix

$$\mathbf{V} = \frac{1}{\gamma} \mathbf{H} \tilde{\mathbf{V}}, \tag{16}$$

where the elements of $\tilde{V}$ are given by $\tilde{\mathbf{V}}_{ij} = -k_i \delta_{ij}$. The noise matrix $\mathbf{F}$ is obtained from the Cholesky decomposition of twice the diffusion matrix $\mathbf{D}$. It is not clear how to define a system with two temperatures and hydrodynamic interactions, since temperature implies the fluctuation-dissipation relation and with hydrodynamic interactions, it is non-local. Nonetheless, if we try to relate this to masses and springs, there are multiple ways to define $\mathbf{D}$, and we consider the following choice,

$$\mathbf{D} = \frac{k_B}{\gamma} \begin{pmatrix} T + \Delta T & J(T + \Delta T) \\ J(T + \Delta T) & T + J^2 \Delta T \end{pmatrix}, \tag{17}$$

where $J = \frac{3d}{4r}$ is the dimensionless parameter quantifying the distance between the particles.

As detailed in Appendix C, following the prescription of Section 5 we arrive at the detailed balance matrix

$$\mathbf{B} = \frac{J(1 - J^2) k_2 k_B \Delta T}{\gamma^2} \begin{pmatrix} 0 & 1 \\ -1 & 0 \end{pmatrix}, \tag{18}$$

and the AER matrix

$$\mathbf{A} = \frac{J(1 - J^2) k_2 k_B \Delta T}{(k_1 + k_2)\gamma} \begin{pmatrix} 0 & 1 \\ -1 & 0 \end{pmatrix}. \tag{19}$$

Note that here and in what follows we apply the prescription in Section 5 assuming that $J$ is constant. This assumption is valid only when $r \gg \sqrt{\langle \Delta x^2 \rangle}$. This is always true in the cases we consider because $\sqrt{\langle \Delta x^2 \rangle} \approx b_0$. The largest driving strength we consider, i.e. $b_0 = 110$ nm is much smaller than the shortest distance between the trap positions, i.e. $r = 2d = 3$ $\mu$m. Moreover, we do not assume $J$ to be fixed in the simulations, yet the theoretical results with fixed $J$ agree remarkably well with the simulation results. This *a posteriori* justifies the assumption in the theory that $J$ is fixed.

The AER scales linearly with the temperature difference $\Delta T$, as in the mass-spring model discussed above. Crucially, for hydrodynamic interactions, to leading order the AER scales linearly with $J$ and hence as $1/r$, as we observe in experiments and simulations. The $\Delta T$ scaling is expected and identical to that obtained for the corresponding mass-spring model as seen

in Eq. (13) [4]. However, the algebraic decay with distance resulting from the hydrodynamic interactions is different from the springs model, for which there is no dependence on particle separation.

# 7 Theory for optically-driven colloidal particles

In this section we consider a system of two hydrodynamically interacting particles where particle 1 experiences nonequilibrium driving which results from the stochastic repositioning of its trap as depicted in Fig. 8b, and thus mimics our experimental set up. This is different from the situation analyzed above, which considered only white noise. Due to the repositioning of the trap, the noise in our experiments is colored, and we employ the prescription discussed in Ref. [40], as outlined below. We consider the one-dimensional motion of two particles, and write the Langevin equation of motion as

$$\frac{d\vec{\delta x}}{dt} = \mathbf{V}\vec{\delta x} + \frac{1}{\gamma}\mathbf{H}\vec{f}. \tag{20}$$

For two particles $\vec{\delta x} = \begin{pmatrix} \delta x_1 \\ \delta x_2 \end{pmatrix}$ is the $\delta x$-displacements of the particles, $\mathbf{V} = -\frac{1}{\gamma}\mathbf{H}\begin{pmatrix} k_1 & 0 \\ 0 & k_2 \end{pmatrix}$ sets the deterministic force applied on each particle, $\mathbf{H} = \begin{pmatrix} 1 & J \\ J & 1 \end{pmatrix}$ is the mobility matrix relating the force on each particle to the velocity of each particle, with $J = \frac{3d}{4r}$ the dimensionless strength of the hydrodynamic interactions. The stochastic active force acting on the particles is $\vec{f} = \begin{pmatrix} f_a(t) \\ 0 \end{pmatrix}$, only the first element of which is non-zero since only the trap of particle 1 is repositioned.

The system also experiences thermal fluctuations, but they obey detailed balance, and thus do not generate probability currents in phase space. Moreover, these fluctuations are uncorrelated with the active forces that arise from trap repositioning, thus there are no probability currents resulting from the interaction between the thermal fluctuations and the active fluctuations. Therefore, for calculating the AER, we consider only the fluctuations resulting from the active force, and do not include thermal fluctuations in the analysis. This is similar to the results presented above for white noise, where the AER depends only on the temperature difference $\Delta T$ and not on the ambient temperature $T$.

The active driving force is the deterministic force pulling particle 1 toward its stochastically varying trap position, $x_{\text{trap}}(t)$, which is updated at time intervals $\tau$. The total force on particle 1 at time $t$ is $-k_1[x(t) - x_{\text{trap}}(t)]$. The first term, $-k_1 x(t)$ is included in the first, deterministic term $\mathbf{V}\vec{\delta x}$ in Eq. (20), while the second term, $k_1 x_{\text{trap}}(t)$ is the stochastic active force that appears in the second term $\frac{1}{\gamma}\mathbf{H}\vec{f}$ in Eq. (20). Thus we identify the active force as $f_a(t) = k_1 x_{\text{trap}}(t)$. Consider two arbitrary times, $t_1 \leq t_2$ along the overall time evolution of the system, measured from the beginning of the experiment. We have non-zero contribution to the force correlation function only if $t_2$ is before the next repositioning event, namely only for $t_2 - t_1 + t < \tau$. The two-time correlation function of the active force is thus

$$\langle f_a(t_1)f_a(t_2)\rangle = \int_0^\tau \frac{dt}{\tau}\theta(s+t-\tau)f(t_1)f(t_2) = \frac{k_1^2}{\tau}\int_0^{\tau-s} dt\langle x_{\text{trap}}^2\rangle = k_1^2 b_0^2\left(1 - \frac{s}{\tau}\right), \tag{21}$$

where $s = t_2 - t_1 \geq 0$.

Comparing Eq. (20) to Eq. (1), we identify $f_a(t) = \sqrt{\gamma k_1} b_0 \xi(t)$, and therefore the noise correlation is $\langle \vec{\xi}(t_1)\vec{\xi}(t_2) \rangle = \mathbb{1}G(s)$, where

$$G(s) = \frac{k_1}{\gamma}\left(1 - \frac{s}{\tau}\right), \quad 0 \le s \le \tau. \tag{22}$$

Considering only the non-equilibrium part, in Eq. (1) the lower triangular noise matrix is given by

$$\mathbf{F} = \sqrt{\frac{k_1}{\gamma}} b_0 \begin{pmatrix} 1 & 0 \\ J & 0 \end{pmatrix}, \tag{23}$$

from which we obtain the diffusion matrix

$$\mathbf{D} = \frac{k_1 b_0^2}{2\gamma} \begin{pmatrix} 1 & J \\ J & J^2 \end{pmatrix}. \tag{24}$$

## 7.1 Infinite imaging rate

Following the colored noise analysis in Ref. [40], the spreading matrix $\mathbf{S}(s)$ is generally defined as

$$\mathbf{S}(s) = 2 \int_0^\infty dt\, e^{\mathbf{V}t} G(t+s). \tag{25}$$

Upon using Eq. (22) for $G(s)$, we obtain for $s \ge 0$

$$\mathbf{S}(s) = \frac{2k_1}{\gamma\tau}\left[\mathbf{V}^{-1}e^{(\mathbf{V}(\tau-s))}\mathbf{V}^{-1} - (\tau - s)\mathbf{V}^{-1} - (\mathbf{V}^{-1})^2\right]. \tag{26}$$

Solving the Lyapunov equation using Eq. (C.3) for the drift matrix and Eq. (24) for the diffusion matrix, we obtain the equal-time white-noise equivalent covariance matrix $\mathbf{C}_w$ as

$$\mathbf{C}_w = \frac{b_0^2}{2(k_1+k_2)} \begin{pmatrix} k_1 + k_2(1-J^2) & k_1 J \\ k_1 J & k_1 J^2 \end{pmatrix}. \tag{27}$$

Finally, using the spreading matrix at time $s = 0$, from Eq. (26) we obtain the AER as [40]

$$\mathbf{A} = \frac{1}{2}\left[\mathbf{S}\mathbf{V}\mathbf{C}_w - \mathbf{C}_w \mathbf{V}^\mathsf{T}\mathbf{S}^\mathsf{T}\right]. \tag{28}$$

Up to leading order in $J$, the AER reads

$$\mathbf{A} = \frac{Jb_0^2 k_1}{(k_1+k_2)\tau}\left[1 + \frac{k_2\exp\left(-\frac{k_1\tau}{\gamma}\right) - k_1\exp\left(-\frac{k_2\tau}{\gamma}\right)}{k_1 - k_2}\right]\begin{pmatrix} 0 & 1 \\ -1 & 0 \end{pmatrix}. \tag{29}$$

 The scaling of the AER as seen in Eq. (29) can be intuitively expected because given a typical displacement $b_0$ of particle 1, the typical displacement of particle 2 resulting from hydrodynamic coupling should be $Jb_0$. Then the area is given by the product of the two displacements and thus $|\mathbf{A}_{12}| \propto Jb_o^2$. Figure 9 shows how our simulations with fast imaging perfectly agree with this theoretical result of the AER. We choose $k_1 = 10$ pN/$\mu$m, $k_2 = 4$ pN/$\mu$m, $\gamma = 0.0126$ pN·s/$\mu$m and vary $\tau$ in the simulations. Figure 9, therefore, describes the effect of varying the trap repositioning rate $1/\tau$ on the AER. The AER peaks close to $k_1\tau/\gamma = 1$, i.e. when the relaxation time $\gamma/k_1$ is comparable to the trap repositioning time $\tau$. In the subsequent sections we present several results for the AER, albeit restricted to the case $k_1\tau/\gamma \gg 1$, i.e. for slow repositioning which is relevant to our experiments. In our experimental set-up, $1/\tau = 36$ Hz, and $k_1 = 2$ pN/$\mu$m, thus $k_1\tau/\gamma = 4.4$. In the subsequent simulations we choose $1/\tau = 36$ Hz, same as in our experimental set-up. To ensure that all subsequent simulations are in the slow repositioning limit, we will use $k_1 = 10$ pN/$\mu$m, for which $k_1\tau/\gamma = 22$.

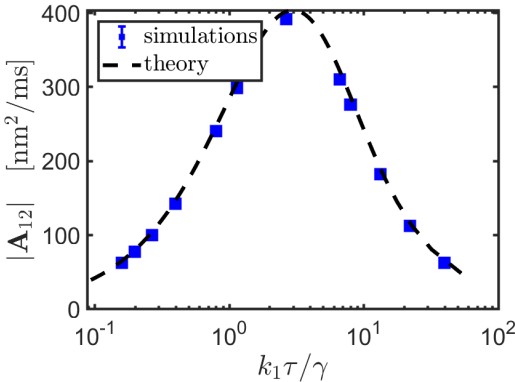

Figure 9: AER vs. $k_1\tau/\gamma$ for two particles separated by a distance of $r = 2d$, and a driving strength of $b_0 = 110$ nm. Simulations at very fast imaging rate of $10^4$ fps are compared with the theoretical Eq. (29). The simulations are with fixed $k_1 = 10$ pN/$\mu$m, $k_2 = 4$ pN/$\mu$m and $\gamma = 0.0126$ pN·s/$\mu$m while $\tau$ is varied to get different values of $k_1\tau/\gamma$.

## 7.2 Finite imaging rate

In Fig. 10a we show the AER as a function of the driving amplitude with a fixed average separation of $r = 2d$ between the particles. The simulation results at a high imaging rate of $10^4$ fps agree very well with the analytical expression given by Eq. (29). Interestingly, we see that the measured AER is significantly smaller in simulations with the experimental imaging rate of 120 fps. This is because a lower imaging rate corresponds to temporal coarse-graining in phase space, thereby reducing the measured area. Figure 12 in Appendix A shows how the AER increases with imaging rate to eventually saturate for fast imaging. Figure 10b, which plots the AER as a function of average distance between the particles for a fixed driving amplitude of $b_0 = 110$ nm, exhibits the same effect of the imaging rate. It also shows excellent agreement between the theoretical prediction and the results from simulations at high imaging rate of $10^4$ fps.

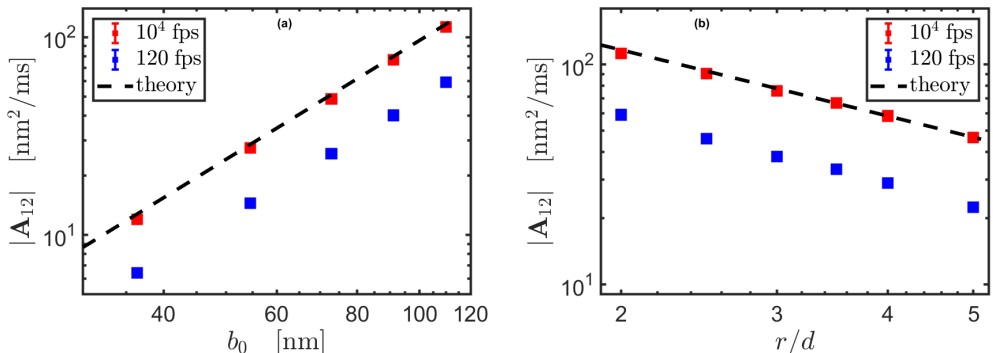

Figure 10: Simulation results of AER for a system of two particles with hydrodynamic interactions. (a) AER vs. driving at interparticle distance $r = 2d$, (b) AER vs. distance at driving strength $b_0 = 110$ nm. Dashed line is Eq. (29), and simulation results are shown for two imaging rates, as indicated in the legend.

# 8 AER for a pair of non-driven particles when a third particle is driven

So far we considered the direct effect of driving. Namely, we probed the AER between a driven particle and another particle, which responds to the driving via the hydrodynamic interactions between them. In extended systems, one expects nonequilibrium fluctuations to propagate. The minimal system for studying this propagation is of three particles, where the first particle is driven, and we measure the AER in the phase space projection of the other two particles. In this section we address the question whether the AER computed from non-driven particles can detect non-equilibrium signatures of a system where there may be untracked driven particles. This is specifically crucial for biological systems where it is not always possible to track all degrees of freedom.

Let us first consider a system of three particles, where particle 1 is in contact with a heat bath at temperature $T + \Delta T$, which is different from the temperature $T$ of the heat baths that particles 2 and 3 are connected to. As discussed in the previous section, the AER depends only on the temperature difference $\Delta T$, and therefore in what follows we set $T = 0$. The particles are in optical traps of generally different stiffnesses $k_j$, and interact hydrodynamically.

We present the drift, diffusion, noise and the covariance matrices in Appendix D. The resultant detailed-balance matrix has non-diagonal elements given by

$$\mathbf{B}_{12} = \frac{J\left[4k_2 + 2Jk_3 - J^2(4k_2 + k_3)\right]k_B\Delta T}{4\gamma^2}, \tag{30a}$$

$$\mathbf{B}_{13} = \frac{J\left[4k_3 + 8Jk_2 - J^2(4k_2 + k_3)\right]k_B\Delta T}{8\gamma^2}, \tag{30b}$$

$$\mathbf{B}_{23} = \frac{J^2\left[2(k_3 - k_2) + J(4k_2 + k_3)\right]k_B\Delta T}{4\gamma^2}. \tag{30c}$$

Note that for three particles connected with springs (see Appendix B), $\mathbf{B}_{23} = 0$, while here $\mathbf{B}_{23} \neq 0$.

Keeping terms up to the lowest order in $J$, the non-diagonal elements of the AER are obtained as

$$\mathbf{A}_{12} = \frac{Jk_2 k_B\Delta T}{\gamma(k_1 + k_2)}, \tag{31a}$$

$$\mathbf{A}_{13} = \frac{Jk_3 k_B\Delta T}{2\gamma(k_1 + k_3)}, \tag{31b}$$

$$\mathbf{A}_{23} = \frac{J^2\left[2(k_3 - k_2) + J(4k_2 - k_3)\right](k_1 k_2 + k_1 k_3 + k_2 k_3)k_B\Delta T}{4\gamma(k_1 + k_2)(k_1 + k_3)(k_2 + k_3)}. \tag{31c}$$

The full expressions for arbitrary $J$ are given in Appendix D. As expected, the AER is proportional to the temperature difference $\Delta T$ between the heat baths. The AER $\mathbf{A}_{12}$ in the subspace of particle 1 and particle 2, and $\mathbf{A}_{13}$ in the subspace of particle 1 and particle 3 are inversely proportional to the distance between the particles. Equation (31c) gives non-zero AER $\mathbf{A}_{23}$ also in the subspace of the non-driven particles, namely, particle 2 and particle 3, while only particle 1 is driven. The AER in this subspace decays faster with distance than in the subspace of driven–non-driven pairs. Interestingly, when $k_2 \neq k_3$ it decays as $1/r^2$ but when $k_2 = k_3$ it decays as $1/r^3$ in the leading order in $J$. The detailed-balance matrix $\mathbf{B}$, as seen from Eq. (30) exhibits the same scaling behaviour with $J$ and $\Delta T$.

Let us now consider the experimentally relevant case, namely, a system of three particles, with particle 1 driven by stochastically repositioning its trap, and where we follow the motion of all three particles along the line connecting them. We could not obtain closed form

expressions of the AER for this case, and in what follows, we use the matrix equation (28) to numerically obtain the exact values of the AER for any set of parameter values.

Within the framework presented in Section 7, i.e. with hydrodynamic interactions and colored noise driving, and considering only the non-equilibrium contributions, the lower triangular noise matrix is given by

$$\mathbf{F} = \sqrt{\frac{k_1}{\gamma}} b_0 \begin{pmatrix} 1 & 0 & 0 \\ J & 0 & 0 \\ \frac{J}{2} & 0 & 0 \end{pmatrix}, \tag{32}$$

from which the diffusion matrix is

$$\mathbf{D} = \frac{k_1 b_0^2}{2\gamma} \begin{pmatrix} 1 & J & \frac{J}{2} \\ J & J^2 & \frac{J^2}{2} \\ \frac{J}{2} & \frac{J^2}{2} & \frac{J^2}{4} \end{pmatrix}, \tag{33}$$

and the drift matrix is given by

$$\mathbf{V} = -\frac{1}{\gamma} \begin{pmatrix} k_1 & J k_2 & \frac{J}{2} k_3 \\ J k_1 & k_2 & J k_3 \\ \frac{J}{2} k_1 & J k_2 & k_3 \end{pmatrix}. \tag{34}$$

Solving the Lyapunov equation using Eq. (34) for the drift matrix and Eq. (33) for the diffusion matrix, we obtain the elements of the equal-time white-noise equivalent covariance matrix $\mathbf{C}_w$, up to leading order in $J$, as

$$(\mathbf{C}_w)_{11} = \frac{b_0^2 \left[ 4k_1 k_2 + 4k_1 k_3 + 4k_2 k_3 + 4k_1^2 - J^2(4k_1 k_2 + k_1 k_3 + 5k_2 k_3) \right]}{8(k_1 + k_2)(k_1 + k_3)}, \tag{35a}$$

$$(\mathbf{C}_w)_{12} = (\mathbf{C}_w)_{21} = \frac{J b_0^2 k_1}{2(k_1 + k_2)}, \tag{35b}$$

$$(\mathbf{C}_w)_{13} = (\mathbf{C}_w)_{31} = \frac{J b_0^2 k_1}{4(k_1 + k_3)}, \tag{35c}$$

$$(\mathbf{C}_w)_{22} = \frac{J^2 b_0^2 k_1}{2(k_1 + k_2)}, \tag{35d}$$

$$(\mathbf{C}_w)_{23} = (\mathbf{C}_w)_{32} = \frac{J^2 b_0^2 k_1(k_1 k_2 + k_1 k_3 + 2k_2 k_3)}{4(k_1 + k_2)(k_1 + k_3)(k_2 + k_3)}, \tag{35e}$$

$$(\mathbf{C}_w)_{33} = \frac{J^2 b_0^2 k_1}{8(k_1 + k_3)}. \tag{35f}$$

The full expressions for arbitrary $J$ are given in Appendix E. We follow the procedure for colored noise, as described in Section 7, to obtain the theoretically predicted AER using Eq. (28).

We simulate a system of three particles, with particle 1 driven along the $x$-axis as before, and we measure the AER between all pairs of particles using a high imaging rate of $10^4$ fps. Figure 11a shows the AER as a function of driving amplitude for all pairs of particles. The simulations were performed with an average distance of $r = 2d$ between each pair of neighbouring particles. A repositioning rate of $1/\tau = 36$ Hz was used, while the trap stiffnesses were $k_1 = 10$ pN/$\mu$m, $k_2 = 4$ pN/$\mu$m, $k_3 = 5$ pN/$\mu$m. The simulation results show a clear scaling of AER with $b_0^2$ in agreement with the theoretical predictions according to Eq. (28). Moreover, noting that particle 1 is driven and that the average distance between particle 1 and particle 3 is twice that between particle 1 and particle 2, we see that $\mathbf{A}_{13}$ is smaller than $\mathbf{A}_{12}$

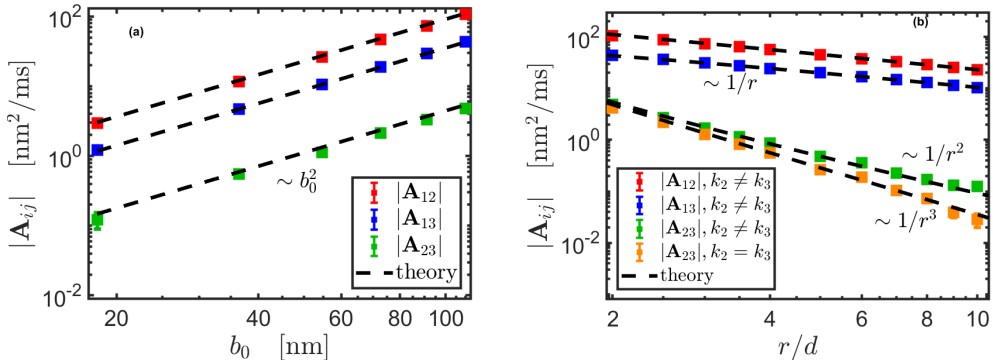

Figure 11: AER for a system of three particles in a harmonic potential with hydro-dynamic interactions between the particles. (a) AER vs. driving for $r = 2d$, (b) AER vs. distance for $b_0 = 110$ nm.

because of the $1/r$ dependence due to hydrodynamic interactions. Remarkably, we observe non-zero AER $\mathbf{A}_{23}$ also for the non-driven pair of particles, albeit the values are much smaller than for the driven-non-driven pairs. This non-zero AER for the non-driven pair of particles is too small to be detected in the experiments, but the simulations clearly exhibit it.

Figure 11b shows the AER as a function of average distance between neighboring particles for all pairs of particles. The simulations were performed with a fixed driving amplitude of $b_0 = 110$ nm. Consider the results from the simulations with the stiffnesses same as in Fig. 11a. These results are labelled $k_2 \neq k_3$ in Fig. 11b. We observe that the measured AER between the driven-non-driven pairs follow the scaling of $\mathbf{A}_{12}, \mathbf{A}_{13} \propto 1/r$, while $\mathbf{A}_{23} \propto 1/r^2$ which agree very well with the theoretical predictions according to Eq. (28), and is the same as in the two-temperature case, as given by Eq. (31). Interestingly, when $k_2 = k_3$, as with the two-temperature case, $\mathbf{A}_{23} \propto 1/r^3$, that is, it decays much faster than when the stiffnesses are unequal. This is exactly what we observe in the simulation results labeled $k_2 = k_3$ in Fig. 11b, where we chose $k_2 = k_3 = 4$ pN/$\mu$m. This fast decay with distance makes it difficult to experimentally detect the non-zero AER for non-driven pairs of particles.

Despite this agreement in the functional dependence on distance in the two-temperature case and in the experimental driving protocol, the ratios between the prefactors of $r$ in $\mathbf{A}$ are different for the two cases. From Eqs. (31a) and (31b) we see that for the case of heat baths at different temperatures $\frac{\mathbf{A}_{12}}{\mathbf{A}_{13}} = \frac{k_2(k_1+k_3)}{k_3(k_1+k_2)}$. With $r = 8d$ – which is large enough such that the leading order in $J$ gives the dominant contribution to the AER – and the other parameters same as used in the simulations presented in Fig. 11b this leads to $\frac{\mathbf{A}_{12}}{\mathbf{A}_{13}} = 1.7$ which differs from $\frac{\mathbf{A}_{12}}{\mathbf{A}_{13}} = 2.2$ resulting from numerically evaluating the colored noise case given by Eq. (28). Similarly, $\frac{\mathbf{A}_{12}}{\mathbf{A}_{23}} = \frac{4k_2(k_1+k_3)(k_2+k_3)}{J(k_1k_2+k_1k_3+k_2k_3)(2(k_3-k_2)+J(4k_2-k_3))}$ which is obtained using Eqs. (31a) and (31c). Again with $r = 8d$ and the other parameters same as used in the simulations presented in Fig. 11b this leads in case of $k_2 \neq k_3$ to $J \cdot \frac{\mathbf{A}_{12}}{\mathbf{A}_{23}} = 6.5$ whereas numerically evaluating the colored noise case given by Eq. (28) results in $J \cdot \frac{\mathbf{A}_{12}}{\mathbf{A}_{23}} = 17.5$. In the case of $k_2 = k_3$ we obtain for the two temperature case $J^2 \cdot \frac{\mathbf{A}_{12}}{\mathbf{A}_{23}} = 1.6$ which is different from $J^2 \cdot \frac{\mathbf{A}_{12}}{\mathbf{A}_{23}} = 3.7$ in the colored noise case.

## 9 Discussion

We have studied the AER to quantify probability currents in a system of hydrodynamically coupled colloidal particles, in which one particle is optically driven. Using this model system, we could identify and decouple the contributions of different experimental parameters, such as driving strength and frequency, interparticle distance, and imaging rate.

We found that due to hydrodynamic interactions between the particles, the AER decays algebraically with inter-particle separation; this contrasts with the fixed AER in elastic systems with local driving. It is, therefore, essential to understand the nature of coupling of the tracked degrees of freedom which are used to measure the AER. For example, a stronger signal would be obtained from objects that are directly connected. Tracer particles attached directly to a biopolymer network, such as actin, would report on motor activity, such as myosin, significantly better and to much larger distances than if embedded in the fluid. We also demonstrate that the AER peaks when the driving time scale $\tau$ is comparable to the relaxation time scale $\gamma/k$. This result is in accord with previous work [23] showing that heat dissipation, another measure of distance from thermal equilibrium, peaks under similar conditions. This result implies that if driving frequency (or relaxation time) can be tuned in the system, one may be able to extract the typical relaxation time (or driving frequency), or alternatively enhance the AER measurement signal in this manner. Another method to ensure proper measurement of the AER is to use an imaging rate that is fast enough compared to the driving and the relaxation time scales.

It is interesting to note that driving one particle can generate probability currents among other non-driven particles. This means that a single active agent can propagate its activity within a series of interacting objects via probability currents. The theoretical approach used in our analysis to calculate the expected AER due to hydrodynamic interactions can be adapted to other interaction types and to larger numbers of particles. This tool could serve as a means to design a system that propagates activity via probability currents in an optimal manner.

Here we focused on the AER as an obervable to detect and quantify non-equilibrium dynamics. It is closely related to the cycling frequency and the entropy production rate which have also been employed as non-equilibrium measures [16, 18]. The cycling frequency is defined as the rate at which a trajectory revolves in coordinate space and its elements are given by [18]

$$\omega_{ij} = \frac{1}{2}\frac{\left(\mathbf{VC} - \mathbf{CV}^{\mathrm{T}}\right)_{ij}}{\sqrt{\det(\mathbf{C}_{[i,j]})}} = \frac{\mathbf{A}_{ij}}{\sqrt{\det(\mathbf{C}_{[i,j]})}}, \tag{36}$$

where $\mathbf{C}_{[i,j]}$ is a $2 \times 2$ matrix with elements $\{\{\mathbf{C}_{ii}, \mathbf{C}_{ij}\}, \{\mathbf{C}_{ji}, \mathbf{C}_{jj}\}\}$. Thus the cycling frequency differs from the AER only by the normalization factor given by the determinant of the Covariance matrix. The entropy production rate (EPR), on the other hand, for a linear system is related to the AER via the relation [16]

$$\mathrm{EPR} = Tr(\mathbf{AC}^{-1}\mathbf{A}^{\mathrm{T}}\mathbf{D}^{-1}). \tag{37}$$

The advantage that the AER and the cycling frequency have over the EPR is due to the fact that they can be computed directly from the raw single particle tracking data. Moreover they can be computed for any two degrees of freedom, while the EPR requires measuring all the degrees of freedom in the system. Indeed the AER can be leveraged, in case of multidimensional systems, to perform a dissipative component analysis to identify the components which contribute the most to EPR, and provide lower bounds on the EPR [16].

We showed that the AER can be a useful observable to detect signatures of non-equilibrium dynamics in a system of two or more particles. In case of a single particle, if it is driven such that it exhibits directional motion (say if the trap driving it moves in circles, rather than along a line) then there would clearly be an observable current in physical space, and we won't

need to search for probability currents in phase space. However, for single-particle systems, if currents are noisy and hard to observe, in principle, one can still detect and quantify them with the AER in physical space. Indeed, Ref. [13] considered probability fluxes to quantify nonequilibrium motion of a beating flagellum of Chlamydomonas reinhardtii by decomposing its motion into different modes.

## Acknowledgments

We thank Shlomi Reuveni for the useful discussions.

**Funding information**  We acknowledge support from the Joint Research Projects on Biophysics Ludwig-Maximilians-Universität München (LMU) - Tel Aviv University (TAU) initiative. YR and DZ acknowledge funding from the Israeli Science Foundation (grant no. 385/21). CB acknowledges financial support from the German Science Foundation (DFG, grant no. 418389167). ST acknowledges support in the form of a Sackler postdoctoral fellowship and funding from the Pikovsky-Valazzi matching scholarship, Tel Aviv University.

## A  AER dependence on frame acquisition rate

Figure 12 shows how the AER varies with the frame acquisition rate, and highlights that at small frame rate the estimated AER values are lower than the steady state value and only for fast imaging do the estimated AER values converge. This is because a lower imaging rate corresponds to temporal coarse-graining in phase space, thereby reducing the measured area.

## B  AER for three particles connected with springs

We consider three particles, where particle 1 is in contact with a heat bath at temperature $T + \Delta T$, which is different from the temperature $T$ of the heat bath that particles 2 and 3 are connected to. Again, we set $T = 0$. The particles themselves are connected to each other and to rigid walls at the ends via springs with spring constants $k_j$. This is an extension of the two-particle case considered in Ref. [4].

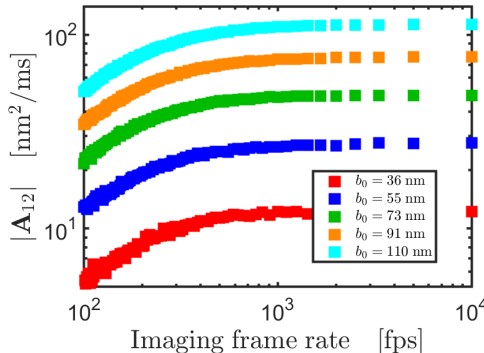

Figure 12: AER vs. imaging frame rate for two particles separated by a distance of $r = 2d$ and different driving strength $b_0$ as shown in the legend.

The matrix $\mathbf{V}$ is given by

$$\mathbf{V} = \frac{1}{\gamma}\begin{pmatrix} -(k_1+k_2) & k_2 & 0 \\ k_2 & -(k_2+k_3) & k_3 \\ 0 & k_3 & -(k_3+k_4) \end{pmatrix}, \tag{B.1}$$

while for the diffusion matrix we choose the form

$$\mathbf{D} = \frac{k_B}{\gamma}\begin{pmatrix} \Delta T & 0 & 0 \\ 0 & 0 & 0 \\ 0 & 0 & 0 \end{pmatrix}, \tag{B.2}$$

and write the noise matrix as

$$\mathbf{F} = \sqrt{\frac{2k_B}{\gamma}}\begin{pmatrix} \sqrt{\Delta T} & 0 & 0 \\ 0 & 0 & 0 \\ 0 & 0 & 0 \end{pmatrix}. \tag{B.3}$$

Solving the Lyapunov equation, we obtain the elements of the covariance matrix,

$$
\begin{aligned}
\mathbf{C}_{11} = \frac{k_B\Delta T}{Z_3}\big(&2k_2^2k_3^3 + 2k_2^3k_3^2 + k_2^2k_4^3 + 2k_2^3k_4^2 + k_3^2k_4^3 + 2k_3^3k_4^2 + 4k_1k_2k_3^3 + k_1k_2^3k_3 \\
&+k_1k_2k_4^3 + k_1k_3^3k_4 + k_1k_3k_4^3 + 4k_1k_3^3k_4 + 3k_2k_3k_4^3 + 8k_2k_3^3k_4 + 5k_2^3k_3k_4 \\
&+6k_1k_2^2k_3^2 + 2k_1^2k_2k_3^2 + k_1^2k_2^2k_3 + 3k_1k_2^2k_4^2 + k_1^2k_2k_4^2 + k_1^2k_2^2k_4 + 4k_1k_3^2k_4^2 \\
&+k_1^2k_3k_4^2 + 2k_1^2k_3^2k_4 + 11k_2k_3^2k_4^2 + 10k_2^2k_3k_4^2 + 14k_2^2k_3^2k_4 + 9k_1k_2k_3k_4^2 \\
&+16k_1k_2k_3^2k_4 + 11k_1k_2^2k_3k_4 + 4k_1^2k_2k_3k_4\big), \tag{B.4a}
\end{aligned}
$$

$$
\begin{aligned}
\mathbf{C}_{12} = \frac{k_2k_B\Delta T}{Z_3}\big(&2k_2^2k_3^2 + 2k_2^2k_4^2 + 3k_3^2k_4^2 + 2k_2k_3^3 + k_2k_4^3 + k_3k_4^3 + 2k_3^3k_4 + 2k_1k_2k_3^2 \\
&+k_1k_2^2k_3 + k_1k_2k_4^2 + k_1k_2^2k_4 + k_1k_3k_4^2 + 2k_1k_3^2k_4 + 6k_2k_3k_4^2 + 8k_2k_3^2k_4 \\
&+5k_2^2k_3k_4 + 4k_1k_2k_3k_4\big), \tag{B.4b}
\end{aligned}
$$

$$
\mathbf{C}_{13} = \frac{k_2k_3k_B\Delta T}{Z_3}\big(2k_2k_3^2 + 2k_2^2k_3 + k_2k_4^2 + 2k_2^2k_4 + k_3k_4^2 + 2k_3^2k_4 + 4k_2k_3k_4\big), \tag{B.4c}
$$

$$
\begin{aligned}
\mathbf{C}_{22} = \frac{k_B\Delta T}{Z_3}\big(&2k_2^2k_3^3 + 2k_2^3k_3^2 + k_2^2k_4^3 + 2k_2^3k_4^2 + k_1k_2^3k_3 + k_1k_2^3k_4 + 5k_2^3k_3k_4 \\
&+k_1k_2^2k_3^2 + k_1k_2^2k_4^2 + 4k_2^2k_3k_4^2 + 5k_2^2k_3^2k_4 + 3k_1k_2^2k_3k_4\big), \tag{B.4d}
\end{aligned}
$$

$$
\mathbf{C}_{23} = \frac{k_3k_B\Delta T}{Z_3}\big(2k_2^2k_3^2 + k_2^2k_4^2 + 2k_2^3k_3 + 2k_2^3k_4 + k_1k_2^2k_3 + k_1k_2^2k_4 + 3k_2^2k_3k_4\big), \tag{B.4e}
$$

$$
\mathbf{C}_{33} = \frac{k_B\Delta T}{Z_3}\big(2\Delta Tk_2^2k_3^3 + 2k_2^3k_3^2 + k_1k_2^2k_3^2 + +k_2^2k_3^2k_4\big), \tag{B.4f}
$$

where

$$
\begin{aligned}
Z_3 = &6k_1k_2^2k_3^3 + 6k_1k_2^3k_3^2 + 4k_1^2k_2k_3^3 + 2k_1^2k_2^3k_3 + 2k_1^3k_2k_3^2 + k_1^3k_2^2k_3 + 2k_1k_2^2k_4^3 + 4k_1k_2^3k_4^2 \\
&+k_1^2k_2k_4^3 + 2k_1^2k_2^3k_4 + k_1^3k_2k_4^2 + k_1^3k_2^2k_4 + k_1k_3^3k_4^3 + 2k_1k_3^3k_4^2 + k_1^2k_3k_4^3 + 4k_1^2k_3^3k_4 \\
&+k_1^3k_3k_4^2 + 2k_1^3k_3^2k_4 + k_2k_3^2k_4^3 + 2k_2k_3^3k_4^2 + 2k_2^2k_3k_4^3 + 6k_2^2k_3^3k_4 + 4k_2^3k_3k_4^2 + 6k_2^3k_3^2k_4 \\
&+8k_1^2k_2^2k_3^2 + 4k_1^2k_2^2k_4^2 + 4k_1^2k_3^2k_4^2 + 8k_2^2k_3^2k_4^2 + 4k_1k_2k_3k_4^3 + 12k_1k_2k_3^3k_4 + 12k_1k_2^3k_3k_4 \\
&+4k_1^3k_2k_3k_4 + 15k_1k_2k_3^2k_4^2 + 18k_1k_2^2k_3k_4^2 + 28k_1k_2^2k_3^2k_4 + 10k_1^2k_2k_3k_4^2 + 18k_1^2k_2k_3^2k_4 \\
&+15k_1^2k_2^2k_3k_4,
\end{aligned}
$$

and we note that the remaining elements of **C** can be obtained from the symmetry of **C**.

The detailed-balance matrix is obtained as

$$\mathbf{B} = \frac{k_2 k_B \Delta T}{\gamma^2} \begin{pmatrix} 0 & -1 & 0 \\ 1 & 0 & 0 \\ 0 & 0 & 0 \end{pmatrix}, \tag{B.5}$$

and the AER matrix has non-diagonal elements given by

$$\mathbf{A}_{12} = -\frac{k_2 k_B \Delta T}{\gamma Q_1} \left( k_1 k_2 + 2 k_1 k_3 + k_1 k_4 + 3 k_2 k_3 + 2 k_2 k_4 + 3 k_3 k_4 + 2 k_3^2 + k_4^2 \right), \tag{B.6a}$$

$$\mathbf{A}_{13} = -\frac{k_2 k_3 k_B \Delta T}{\gamma Q_1} (k_2 + 2 k_3 + k_4), \tag{B.6b}$$

$$\mathbf{A}_{23} = -\frac{k_2^2 k_3 k_B \Delta T}{\gamma Q_1}, \tag{B.6c}$$

where

$$\begin{aligned}
Q_1 = &\ k_1^2 k_2 + 2 k_1^2 k_3 + k_1^2 k_4 + 2 k_1 k_2^2 + 8 k_1 k_2 k_3 + 4 k_1 k_2 k_4 + 4 k_1 k_3^2 + 4 k_1 k_3 k_4 + k_1 k_4^2 \\
&\ + 6 k_2^2 k_3 + 4 k_2^2 k_4 + 6 k_2 k_3^2 + 8 k_2 k_3 k_4 + 2 k_2 k_4^2 + 2 k_3^2 k_4 + k_3 k_4^2.
\end{aligned}$$

# C  Covariance matrix for a system of two particles in contact with heat baths at different temperatures

We consider the diffusion matrix to be given by

$$\mathbf{D} = \frac{k_B}{\gamma} \begin{pmatrix} T + \Delta T & J(T + \Delta T) \\ J(T + \Delta T) & T + J^2 \Delta T \end{pmatrix}, \tag{C.1}$$

where $J = \frac{3d}{4r}$ is the dimensionless parameter quantifying the distance between the particles. The Cholesky decomposition $\mathbf{D} = \frac{1}{2} \mathbf{F} \mathbf{F}^{\mathrm{T}}$ gives

$$\mathbf{F} = \sqrt{\frac{2 k_B}{\gamma}} \begin{pmatrix} \sqrt{T + \Delta T} & 0 \\ J\sqrt{T + \Delta T} & \sqrt{T(1 - J^2)} \end{pmatrix}. \tag{C.2}$$

The drift matrix is

$$\mathbf{V} = \frac{-1}{\gamma} \begin{pmatrix} k_1 & J k_2 \\ J k_1 & k_2 \end{pmatrix}. \tag{C.3}$$

Solving Eq. (5) gives the covariance matrix, the elements of which are given by

$$\mathbf{C}_{11} = \frac{k_B}{k_1(k_1 + k_2)} \left[ T(k_1 + k_2) + \Delta T(k_1 + k_2 - J^2 k_2) \right], \tag{C.4a}$$

$$\mathbf{C}_{12} = \mathbf{C}_{21} = \frac{J k_B \Delta T}{k_1 + k_2}, \tag{C.4b}$$

$$\mathbf{C}_{22} = \frac{k_B}{k_2(k_1 + k_2)} \left[ T(k_1 + k_2) + J^2 k_2 \Delta T \right]. \tag{C.4c}$$

# D Three colloidal particles driven by temperature difference

The matrix $\mathbf{V}$ is given by

$$\mathbf{V} = \frac{-1}{\gamma} \begin{pmatrix} k_1 & Jk_2 & \frac{J}{2}k_3 \\ Jk_1 & k_2 & Jk_3 \\ \frac{Jk_1}{2} & Jk_2 & k_3 \end{pmatrix}, \tag{D.1}$$

while we consider the diffusion matrix to be given by

$$\mathbf{D} = \frac{k_B}{\gamma} \begin{pmatrix} \Delta T & J\Delta T & \frac{J}{2}\Delta T \\ J\Delta T & J^2\Delta T & \frac{J^2}{2}\Delta T \\ \frac{J}{2}\Delta T & \frac{J^2}{2}\Delta T & \frac{J^2}{4}\Delta T \end{pmatrix}. \tag{D.2}$$

The Cholesky decomposition of $\mathbf{D} = \frac{1}{2}\mathbf{F}\mathbf{F}^{\mathsf{T}}$ gives

$$\mathbf{F} = \sqrt{\frac{2k_B}{\gamma}} \begin{pmatrix} \sqrt{\Delta T} & 0 & 0 \\ J\sqrt{\Delta T} & 0 & 0 \\ \frac{J\sqrt{\Delta T}}{2} & 0 & 0 \end{pmatrix}. \tag{D.3}$$

Solving the Lyapunov equation, we obtain the elements of the covariance matrix $\mathbf{C}$,

$$\begin{aligned}
\mathbf{C}_{11} = \frac{k_B\Delta T}{4k_1 Z_4} \Big[ & 16k_1k_2^2 + 16k_1^2k_2 + 16k_1k_3^2 + 16k_1^2k_3 + 16k_2k_3^2 + 16k_2^2k_3 + 32k_1k_2k_3 \\
& -4J^2(8k_1k_2^2 - 4k_1^2k_2 - 2k_1k_3^2 - k_1^2k_3 - 9k_2k_3^2 - 9k_2^2k_3 - 5k_1k_2k_3) \\
& + 16J^3(k_2k_3^2 + k_2^2k_3 - k_1k_2k_3) + J^4(k_1k_3^2 + 16k_1k_2^2 + 8k_1k_2k_3) \Big],
\end{aligned} \tag{D.4a}$$

$$\begin{aligned}
\mathbf{C}_{12} = \frac{Jk_B\Delta T}{2Z_4} \Big[ & 8k_1k_2 + 8k_1k_3 + 8k_2k_3 + 8k_3^2 - 4J(k_3^2 + k_2k_3) \\
& -2J^2(k_3^2 - 4k_1k_2 + k_1k_3 - k_2k_3) + J^3(k_3^2 - 4k_2k_3) \Big],
\end{aligned} \tag{D.4b}$$

$$\begin{aligned}
\mathbf{C}_{13} = \frac{Jk_B\Delta T}{2Z_4} \Big[ & 4k_1k_2 + 4k_1k_3 + 4k_2k_3 + 4k_2^2 - 8J(k_2^2 + k_2k_3) \\
& + J^2(4k_2k_3 - 4k_2^2 - 4k_1k_2 - k_1k_3) + 2J^3(4k_2^2 - k_2k_3) \Big],
\end{aligned} \tag{D.4c}$$

$$\begin{aligned}
\mathbf{C}_{22} = \frac{J^2 k_B\Delta T}{Z_4} \Big[ & 4k_1k_2 + 4k_1k_3 + 4k_2k_3 + 4k_3^2 - 4Jk_3^2 \\
& + J^2(k_3^2 - 4k_1k_2 - k_1k_3 - 4k_2k_3) \Big],
\end{aligned} \tag{D.4d}$$

$$\mathbf{C}_{23} = \frac{J^2 k_B\Delta T}{2Z_4} \Big[ 4k_1k_2 + 4k_1k_3 + 8k_2k_3 - 10Jk_2k_3 - J^2(4k_1k_2 + k_1k_3) \Big], \tag{D.4e}$$

$$\begin{aligned}
\mathbf{C}_{33} = \frac{J^2 k_B\Delta T}{4Z_4} \Big[ & 4k_1k_2 + 4k_1k_3 + 4k_2k_3 + 4k_2^2 - 16Jk_2^2 \ , \\
& + J^2(16k_2^2 - 4k_1k_2 - k_1k_3 - 4k_2k_3) \Big],
\end{aligned} \tag{D.4f}$$

where

$$\begin{aligned}
Z_4 = & 4k_1k_2^2 + 4k_1^2k_2 + 4k_1k_3^2 + 4k_1^2k_3 + 4k_2k_3^2 + 4k_2^2k_3 - 4J^2k_1k_2^2 - 4J^2k_1^2k_2 - J^2k_1k_3^2 \\
& - J^2k_1^2k_3 - 4J^2k_2k_3^2 - 4J^2k_2^2k_3 + 8k_1k_2k_3 - 4J^3k_1k_2k_3 \ ,
\end{aligned}$$

and we note that the remaining elements of $\mathbf{C}$ can be obtained from the symmetry of $\mathbf{C}$.

The full expressions of the elements of the AER presented in Eq. (31) are

$$
\begin{aligned}
\mathbf{A}_{12} = -\mathbf{A}_{21} = \frac{J k_B \Delta T}{4\gamma Q_2} \Big[ & 16k_2^2 k_3 + 16k_1 k_2^2 + 16k_2 k_3^2 + 16k_1 k_2 k_3 + 8J(k_1 k_2 k_3 + k_1 k_3^2) \\
& -4J^2(6k_1 k_2 k_3 + k_1 k_3^2 + 8k_2^2 k_3 + 8k_1 k_2^2 + 10k_2 k_3^2) + 2J^3(13k_2 k_3^2 - 4k_1 k_2 k_3 - k_1 k_3^2) \\
& +J^4(16k_2^2 k_3 + 16k_1 k_2^2 - 4k_2 k_3^2 + 8k_1 k_2 k_3 + k_1 k_3^2) \Big],
\end{aligned}
\tag{D.5a}
$$

$$
\begin{aligned}
\mathbf{A}_{13} = -\mathbf{A}_{31} = \frac{J k_B \Delta T}{8\gamma Q_2} \Big[ & (2-J)(16J^3 k_2^2 k_3 - 16J^3 k_1 k_2^2 - 4J^3 k_2 k_3^2 - 8J^3 k_1 k_2 k_3 \\
& -J^3 k_1 k_3^2 + 24J^2 k_2^2 k_3 - 8J^2 k_2 k_3^2 - 8J^2 k_1 k_2 k_3 - 2J^2 k_1 k_3^2 + 4J k_2^2 k_3 \\
& + 4J k_2 k_3^2 - 16J k_1 k_2^2 + 20J k_1 k_2 k_3 + 4J k_1 k_3^2 + 8k_2^2 k_3 + 8k_2 k_3^2 + 8k_1 k_2 k_3 + 8k_1 k_3^2) \Big],
\end{aligned}
\tag{D.5b}
$$

$$
\begin{aligned}
\mathbf{A}_{23} = -\mathbf{A}_{32} = -\frac{J^2 k_B \Delta T}{4\gamma Q_2} & (2k_2 - 2k_3 - 4J k_2 + J k_3) \\
& \times (4k_1 k_2 + 4k_1 k_3 + 4k_2 k_3 - 4J^2 k_1 k_2 - J^2 k_1 k_3 - 4J^2 k_2 k_3),
\end{aligned}
\tag{D.5c}
$$

where

$$
\begin{aligned}
Q_2 = & -4J^3 k_1 k_2 k_3 - 4J^2 k_1^2 k_2 - J^2 k_1^2 k_3 - 4J^2 k_1 k_2^2 - J^2 k_1 k_3^2 - 4J^2 k_2^2 k_3 - 4J^2 k_2 k_3^2 + 4k_1^2 k_2 \\
& + 4k_1^2 k_3 + 4k_1 k_2^2 + 8k_1 k_2 k_3 + 4k_1 k_3^2 + 4k_2^2 k_3 + 4k_2 k_3^2.
\end{aligned}
$$

# E  Full expressions of the covariance matrix presented in Eq. (35)

The full expressions of the elements of the equal-time white-noise equivalent covariance matrix $\mathbf{C}_w$ presented in Eq. (35) are

$$
\begin{aligned}
(\mathbf{C}_w)_{11} = \frac{b_0^2}{8Z_2} \big( & 16k_1 k_2^2 + 16k_1^2 k_2 + 16k_1 k_3^2 + 16k_1^2 k_3 + 16k_2 k_3^2 + 16k_2^2 k_3 + 32k_1 k_2 k_3 \\
& -32J^2 k_1 k_2^2 - 16J^2 k_1^2 k_2 - 8J^2 k_1 k_3^2 - 4J^2 k_1^2 k_3 - 36J^2 k_2 k_3^2 - 36J^2 k_2^2 k_3 \\
& -20J^2 k_1 k_2 k_3 + 16J^3 k_2 k_3^2 + 16J^3 k_2^2 k_3 - 16J^3 k_1 k_2 k_3 + J^4 k_1 k_3^2 \\
& +16J^4 k_1 k_2^2 + 8J^4 k_1 k_2 k_3 \big),
\end{aligned}
\tag{E.1a}
$$

$$
\begin{aligned}
(\mathbf{C}_w)_{12} = \frac{J b_0^2 k_1}{4Z_2} \big( & 8k_1 k_2 + 8k_1 k_3 + 8k_2 k_3 + 8k_3^2 - 4J k_3^2 - 4J k_2 k_3 - 2J^2 k_3^2 \\
& -8J^2 k_1 k_2 - 2J^2 k_1 k_3 + 2J^2 k_2 k_3 + J^3 k_3^2 - 4J^3 k_2 k_3 \big),
\end{aligned}
\tag{E.1b}
$$

$$
\begin{aligned}
(\mathbf{C}_w)_{13} = \frac{J b_0^2 k_1}{4Z_2} \big( & 4k_1 k_2 + 4k_1 k_3 + 4k_2 k_3 - 8J k_2^2 + 4k_2^2 - 4J^2 k_2^2 + 8J^3 k_2^2 - 8J k_2 k_3 \\
& -4J^2 k_1 k_2 - J^2 k_1 k_3 + 4J^2 k_2 k_3 - 2J^3 k_2 k_3 \big),
\end{aligned}
\tag{E.1c}
$$

$$
\begin{aligned}
(\mathbf{C}_w)_{22} = \frac{J^2 b_0^2 k_1}{2Z_2} \big( & 4k_1 k_2 + 4k_1 k_3 + 4k_2 k_3 - 4J k_3^2 + 4k_3^2 + J^2 k_3^2 - 4J^2 k_1 k_2 \\
& -J^2 k_1 k_3 - 4J^2 k_2 k_3 \big),
\end{aligned}
\tag{E.1d}
$$

$$
(\mathbf{C}_w)_{23} = \frac{J^2 b_0^2 k_1}{4Z_2} \big( 4k_1 k_2 + 4k_1 k_3 + 8k_2 k_3 - 10J k_2 k_3 - 4J^2 k_1 k_2 - J^2 k_1 k_3 \big),
\tag{E.1e}
$$

$$
\begin{aligned}
(\mathbf{C}_w)_{33} = \frac{J^2 b_0^2 k_1}{8Z_2} \big( & 4k_1 k_2 + 4k_1 k_3 + 4k_2 k_3 - 16J k_2^2 + 4k_2^2 + 16J^2 k_2^2 - 4J^2 k_1 k_2 \\
& -J^2 k_1 k_3 - 4J^2 k_2 k_3 \big),
\end{aligned}
\tag{E.1f}
$$

where

$$Z_2 = 4k_1 k_2^2 + 4k_1^2 k_2 + 4k_1 k_3^2 + 4k_1^2 k_3 + 4k_2 k_3^2 + 4k_2^2 k_3 - 4J^2 k_1 k_2^2 - 4J^2 k_1^2 k_2 - J^2 k_1 k_3^2$$
$$- J^2 k_1^2 k_3 - 4J^2 k_2 k_3^2 - 4J^2 k_2^2 k_3 + 8k_1 k_2 k_3 - 4J^3 k_1 k_2 k_3 \,,$$

and note that the other elements are given by the symmetric property of $\mathbf{C}_w$.

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
