# Peer review of "Nonequilibrium Probability Currents in Optically-Driven Colloidal Suspensions"

_SciPost Physics, doi:SciPost Phys. 17, 096 (2024)_

## Round 1 · Referee Report · Anonymous (Referee 2) · 2024-7-26

Report

I have read the new version of the manuscript and the rebuttals by the authors. I believe that they have addressed satisfactorily the various points raised. Consequently, I now support publication of the manuscript in Sci Post.

Recommendation

Publish (easily meets expectations and criteria for this Journal; among top 50%)

  • validity: top
  • significance: high
  • originality: top
  • clarity: high
  • formatting: excellent
  • grammar: excellent

Author:  Samudrajit Thapa  on 2024-08-22  [id 4708]

(in reply to Report 1 on 2024-07-26)

We thank the referee for the positive evaluation of our work and for recommending publication.

---

## Round 1 · Referee Report · Anonymous (Referee 3) · 2024-7-29

Report

I'm not entirely satisfied with the reply but I leave the following comments as optional. First, "phase space" is a concept from Hamiltonian mechanics that describes the space spanned by both coordinates and momenta, which is not applicable here. I don't know what "reduced phase space" is supposed to be. I advise to simply drop the term "phase space". Second, I still think performing a Taylor expansion of the mobility tensor should not be too difficult and would be a much stronger result than simply positing that J can be treated as a constant.

Recommendation

Publish (meets expectations and criteria for this Journal)

  • validity: -
  • significance: -
  • originality: -
  • clarity: -
  • formatting: -
  • grammar: -

Author:  Samudrajit Thapa  on 2024-08-22  [id 4710]

(in reply to Report 3 on 2024-07-29)

We thank the referee for recommending publication.

Following the comment on "phase space", we now explicitly state on page 3 of the paper that the entire phase space of the system includes the positions and momenta of all particles, and we clarify that our analysis of the AER is in projections to two-dimensional subspaces of phase space.

The second optional comment suggests performing a Taylor series expansion in J, rather than assuming that it is constant. We provide quantitative arguments that the variations in J are very small, and use them to justify why J can be treated as constant. We stress that the numerical simulations took into account the precise locations of the particles, and did not assume that J is constant. The remarkable agreement between simulations with varying J and the theoretical derivations with fixed J, provide another confirmation for the validity of using fixed J in the theory. We have added this explanation to the revised manuscript on page 17.

---

## Round 1 · Referee Report · Anonymous (Referee 1) · 2024-7-29

Report

The authors have made a good job improving their manuscript by rearranging the different part, solving their issue with Gaussian traps and answering the comments of the Referrees, including my own. I believe the paper now meet the SciPost Physics criteria and I am ready to recommend it soon.

I just have one last suggestion that I think could improve the comparison between experiments and simulations if the authors are willing to consider it.

Main remark

  • In Fig. 4, there is a one-order of magnitude difference between experiments and simulations. The authors argue that one cause may be that hydrodynamic interactions close to a wall are weaker. I wonder whether this point can be made more quantitative. Did the authors try to introduce a modified hydrodynamic tensor to account for the presence of the walls? Does doing so improve the agreement between experiments and simulations? (See for instance [Blake, Proc. Gamb. Phil. Soc. (1971)] or the book [Microhydrodynamics, Kim & Karila (1991)] as general references.)

Typographic points

  • I believe the standard way to print units is in roman font (non-italic, non-math). If the authors use LaTex, they may use the siunitx package.

  • Typos in the introduction (I haven't checked the full text). " fluctuations in them stems [stem] from" " phase space of the system; [.] The"

Recommendation

Ask for minor revision

  • validity: high
  • significance: high
  • originality: high
  • clarity: high
  • formatting: good
  • grammar: good

Author:  Samudrajit Thapa  on 2024-08-22  [id 4709]

(in reply to Report 2 on 2024-07-29)

We thank the referee for the positive remarks and the suggestions.

Regarding the remark on quantifying the effect of the confining walls on the hydrodynamic interactions and the resultant suppression of the AER in the experiments compared to the simulations, indeed the presence of walls leads to weaker hydrodynamic interactions between the particles, and causes it to decay faster with the distance between them as compared to the case without walls (Refs. 35-37 in the revised manuscript). Thus, the motion of each particle due to the motion of other particles is smaller, and this results in a lower AER. However, because of the noise in the experimental data, the lack of precise information on the distance to the walls and the limited range of distance between the particles which can be explored, we are unable to provide a quantitative description of the effect of the walls. We have included this discussion on pages 9 and 10 of the revised manuscript.

We thank the referee for spotting the typographic errors, which we have corrected in the revised manuscript.

We hope that the referee will now deem our work suitable for publication.

---

## Round 1 · Author Response

Dear SciPost Physics Editors,

We are resubmitting our manuscript, "Nonequilibrium Probability Currents in Optically-Driven Colloidal Suspensions".

We sincerely appreciate the referees' recognition of our manuscript's potential and their acknowledgment of its importance as an "insightful minimal model" with "wider application." We are grateful for their positive and thorough review and attentive reading. In response to their feedback, we have comprehensively revised the manuscript. Below, we address the two main comments raised by the referees. More detailed and specific responses to all comments are provided in our enclosed responses to each of the referee reports. In the revised manuscript we highlight in red the modifications that we introduced.

In response to the referees' primary request, we have restructured the paper to clarify its main message and improve its flow. The revised structure is as follows:

  1. We begin by emphasizing the significance of the Area Enclosing Rate (AER), a non-invasive measure of probability currents (Battle et al., Science). We explain its crucial role in identifying departures from equilibrium, particularly in systems where direct evidence of dynamical evolution or currents is absent.
  2. We then highlight that while the primary models developed to understand the relationship between a system's internal activity and the measured AER have focused on systems coupled by elastic springs, many microscopic systems (such as complex fluids and biological matter) are predominantly governed by hydrodynamic interactions.
  3. Following this introduction, we present a minimal physical model incorporating hydrodynamic interactions. We employ a multifaceted approach, combining experiments, theoretical analysis, and numerical simulations to elucidate the relationship between activity and AER.
  4. We conclude with a comprehensive discussion of our model's implications and findings.

This restructured format provides a clear progression from the fundamental concept of AER to the novel contributions of our study, emphasizing the importance of hydrodynamic interactions in non-equilibrium systems.

Another request by the referees was to provide a better connection between the AER and entropy production rate (EPR). We have added a full paragraph to the discussion section. In short, the EPR for a linear system is given by the trace of the matrix computed as a product of the AER, the inverse Covariance matrix, and the inverse Diffusion matrix (see Ref. [15] and Eq. 37 in the revised manuscript). The advantage of the AER over the EPR is that it can be computed directly, for any two degrees of freedom, directly from the raw measured trajectories. In contrast, the EPR requires the measurement of the probability of each trajectory and its time-reversed trajectory, two non-trivial, data-intensive measures.

The significance of our work is twofold. First, we provide a crucial extension of the previous work on the AER by studying its behavior when hydrodynamic interactions dominate particle dynamics. These conditions extend the applicability of the AER as an analysis tool for complex active fluids and biological systems. Significantly, we also show that the AER signal decays rapidly with distance in such systems, in contrast to elastically bound systems. This renders the AER a more local measure than expected in hydrodynamic-interaction-dominated systems.

We thank you for the opportunity to improve our work and hope you find the revised manuscript suitable for publication in SciPost Physics.

Sincerely, Samudrajit Thapa (on behalf of the authors)

---

## Round 2 · Author Response

Dear Editor,

We were happy to see that all three referees recommended or leaned toward recommending the acceptance for publication of our paper in SciPost Physics. As detailed below, we have answered all the remaining minor or optional issues mentioned by the referees. We hope that this will enable the journal to accept the paper, and we will be happy to address any further issues if such would arise.

Referee 2 suggested to consider making more quantitative the effect of the confining walls on the hydrodynamic interactions and the resultant suppression of the AER in the experiments compared to the simulations. The presence of walls leads to weaker hydrodynamic interactions between the particles, and causes it to decay faster with the distance between them as compared to the case without walls (Refs. 35-37 in the revised manuscript). Thus, the motion of each particle due to the motion of other particles is smaller, and this results in a lower AER. However, because of the noise in the experimental data, the lack of precise information on the distance to the walls and the limited range of distance between the particles which can be explored, we are unable to provide a quantitative description of the effect of the walls. We have included this discussion on page 9 of the revised manuscript.

Referee 2 also spotted several typographic errors, which we corrected in the revised manuscript.

Referee 3 made two optional comments. The first is about the term “phase space”. Following this comment, we now explicitly state in page 3 of the paper that the entire phase space of the system includes the positions and momenta of all particles, and we clarify that our analysis of the AER is in projections to two-dimensional subspaces of phase space.

The second optional comment made by Referee 3 suggested performing a Taylor series expansion in J, rather than assuming that it is constant. We provide quantitative arguments that the variations in J are very small, and use them to justify why J can be treated as constant. We stress that the numerical simulations took into account the precise locations of the particles, and did not assume that J is constant. The remarkable agreement between simulations with varying J and the theoretical derivations with fixed J, provide another confirmation for the validity of using fixed J in the theory. We have added this explanation to the revised manuscript on page 17.

Sincerely,
Samudrajit Thapa
on behalf of all authors

---

## Editorial Decision

published